# Effects of the Exopolysaccharide from *Lactiplantibacillus plantarum* HMX2 on the Growth Performance, Immune Response, and Intestinal Microbiota of Juvenile Turbot, *Scophthalmus maximus*

**DOI:** 10.3390/foods12102051

**Published:** 2023-05-19

**Authors:** Gege Hu, Yihui Wang, Rui Xue, Tongji Liu, Zengjia Zhou, Zhennai Yang

**Affiliations:** Key Laboratory of Geriatric Nutrition and Health of Ministry of Education, Beijing Advanced Innovation Center for Food Nutrition and Human Health, Beijing Engineering and Technology Research Center of Food Additives, Beijing Technology and Business University, No. 11 Fu-Cheng Road, Hai-Dian District, Beijing 100048, China; hugege0819@163.com (G.H.); wangyihui202103@163.com (Y.W.); 2130021028@st.btbu.edu.cn (R.X.); 2130021006@st.btbu.edu.cn (T.L.); m15255825668@163.com (Z.Z.)

**Keywords:** *Lactiplantibacillus plantarum*, exopolysaccharide (EPS), immunity, intestinal health, *Scophthalmus maximus*

## Abstract

In this study, the exopolysaccharide (EPS) from *Lactiplantibacillus plantarum* (HMX2) was isolated from Chinese Northeast Sauerkraut. Its effects on juvenile turbot were investigated by adding different concentrations of HMX2-EPS (C: 0 mg/kg, H1: 100 mg/kg, H2: 500 mg/kg) to the feed. Compared with the control group, HMX2-EPS significantly improved the growth performance of juvenile turbot. The activities of antioxidant enzymes, digestive enzymes, and immune-related enzymes were significantly increased. HMX2-EPS could also increase the secretion of inflammatory factors and enhance the immune response of turbot by regulating the IFN signal transduction pathway and exhibit stronger survival rates after the *A. hydrophila* challenge. Moreover, HMX2-EPS could improve the diversity of intestinal microbiota in juvenile fish, increase the abundance of potential probiotics, and reduce the abundance of pathogenic bacteria. The function of gut microbes in metabolism and the immune system could also be improved. All results showed better effects with high concentrations of HMX2-EPS. These results indicated that HMX2-EPS supplementation in the diet could promote growth, improve antioxidant activity, digestive capacity, and immunity capacity, and actively regulate the intestinal microbiota of juvenile turbot. In conclusion, this study might provide basic technical and scientific support for the application of *L. plantarum* in aquatic feed.

## 1. Introduction

Probiotics are widely used in aquaculture because of their diverse beneficial properties, as they have the capability to improve growth performance and antioxidant levels and enhance innate immune response [1]. Probiotics can effectively reduce the number of antibiotics in aquaculture, thereby reducing the production of drug-resistant bacteria, and preventing side effects such as drug residues [2]. Therefore, the use of probiotics on aquatic animals has become a very effective measure to replace antibiotics [3]. Recently, more and more researchers have paid attention to the secondary metabolites of probiotics, such as exopolysaccharide (EPS). EPSs have unique physicochemical properties and biological activities, so they are applied in food, medicine, and other industrial products, which has stimulated interest in exploring and discovering new EPSs [4].

EPS is a high molecular weight long-chain polymer that is mainly produced by the metabolism of microorganisms, including cyanobacteria, fungi, and bacteria [5]. Lactic acid bacteria (LAB) have attracted unique research interest among EPS-producing bacteria due to their beneficial properties of recognized safety [6]. As an important secondary metabolite of LAB, EPS has many biological functions, making it one of the most potential substitutes for immunomodulators. The modulatory effects of LAB-derived EPS on host immunity, antioxidant activity, and intestinal microbiota have been extensively studied. EPS from *Lactobacillus casei* WXD030 could enhance RAW264.7 cell proliferation and phagocytic activity, as well as induce NO, TNF-α, IL-1β, and IL-6 production to play a role in regulating immune responses [7]. EPS produced by *Lactobacillus plantarum* RS20D could stimulate macrophages to release NO and up-regulate gene expression of pro-inflammatory cytokines at the mRNA level [8]. Liu et al. found that EPS from *Lactobacillus* Y42 exerted antioxidant function by up-regulating the expression and activities of antioxidant enzyme lines such as catalase (CAT), superoxide dismutase (SOD), and glutathione peroxidase (GSH-PX) in HT-29 cells [9]. EPS produced by *Lactobacillus plantarum* YW11 effectively recovered the intestinal microbial diversity in mice and increased the abundance of Roseburia, Ruminococcus, and Blautia with an increased content of butyric acid [10]. Many in vitro research results showed that EPS could effectively inhibit the growth of harmful bacteria such as *Enterobacter sakazakii*, *Escherichia coli*, *Listeria monocytogenes*, *Staphylococcus aureus*, *Candida albicans,* and *Salmonella typhimurium* [11]. It has been reported that some EPSs of LAB can be effective stimulants for improving the growth performance and immunity of fish and have a promising application in aquaculture due to their biological activities, such as antibacterial, antioxidant, anticancer, and modulating immunity. Feng et al. found that EPS of *Lactococcus lactis* Z-2 isolated from healthy carp could enhance its immune function, antioxidant capacity, and disease resistance [12]. However, studies on the growth performance, immune response, disease resistance, and intestinal microbiota effects of EPS from food source probiotics on aquatic animals have been reported relatively little.

In this study, *Scophthalmus maximus* is an economically important species of flounder known as turbot. Its meat is delicious and rich in nutritional value, and it is widely cultivated in Europe and China [13]. At present, there is little research that has reported on food-derived probiotics exopolysaccharides as feed additives for aquatic animals. We isolated a strain of *Lactiplantibacillus plantarum* HMX2 that was isolated from Chinese Northeast Sauerkraut. EPS was extracted to investigate the effects on growth performance, antioxidant activities, digestive enzyme activities, immune capacity, and intestinal microbiota of juvenile turbot. The microbiome technology was used to reveal the regulatory mechanism of HMX2-EPS on the composition and diversity of core taxa in turbot to clarify the prebiotic effect of the EPS and provide the basic data for the application of *L. plantarum* in turbot farming. This study could further reveal the role of LAB in aquatic feed, which had positive significance for the comprehensive development and utilization of LAB.

## 2. Materials and Methods

### 2.1. Preparation of EPS Samples

*Lactiplantibacillus plantarum* HMX2 was provided by the Dairy Laboratory at Beijing Technology and Business University of China. HMX2 was cultured in an SDM medium adjusted to pH 6.6 with 1 M acetic acid [14]. The EPS produced by the strain was then isolated and prepared using the method previously reported [15]. A total of 80% (wt/vol) trichloroacetic acid (TCA) (Shanghai Aladdin Biochemical Technology Co., Ltd., Shanghai, China) was added to the fermentation broth to achieve the 4% (wt/vol) trichloroacetic acid concentration. After stirring for 2 h at room temperature, followed by centrifugation at 4 °C for 30 min, 2 volumes of ethanol (Shanghai Aladdin Biochemical Technology Co., Ltd., Shanghai, China) were added to the supernatant and then kept overnight at 4 °C. The pellet of crude HMX2-EPS was collected after centrifugation at 12,000× *g* for 30 min at 4 °C. The precipitate was dialyzed and lyophilized with a lyophilizer (Christ, Hagen, Germany). Then, the protein was removed by the Sevage method, and the pigment was removed by activated carbon [16] to obtain the HMX2-EPS.

### 2.2. Experimental Design and Daily Management

The healthy juvenile *S. maximus* used in the experiment were purchased from a mariculture company in Yantai City, China. The experimental fish were temporarily maintained in aerated artificial seawater for 7 days, siphoning 30% of the water every day. During the temporary cultivation, the water temperature was controlled at (17 ± 1) °C, and the basic feed was fed after changing the water every day (Yantai Penganyuan Marine Food Co., Ltd., Yantai, China). According to our experimental arrangement, 300 juvenile turbot were randomly divided into 3 groups with 100 fish in each group and repeated four times, which were then divided into a control group and two experimental groups. The control group (C) was fed with basic feed, while the experimental group was fed with basic feed supplemented with 100 mg/kg (H1) and 500 mg/kg (H2) HMX2-EPS, respectively, and each group had three replicates. The culture experiment was conducted for 6 weeks, during which the water was changed once a day, with 1/3 to 1/2 water each time. The residual bait and feces were removed, and the health status of the fish was recorded. After the aquaculture experiment, the experimental fish were fasted for 24 h. All fish in the tank were fished out, and the quantity, length, and weight were recorded, respectively.

### 2.3. Sample Collection

Ten fish were randomly collected, and the intestines were quickly dissected and separated with 100 mg/L MS 222 (Beijing Green Hengxing Biotechnology Co., Ltd., Beijing, China). The intestines were washed with sterilized 0.85% normal saline and placed in RNA-free centrifuge tubes and frozen storage tubes and placed in liquid nitrogen. The intestines were then placed in an RNA-free centrifuge tube and a cryopreservation tube and placed in liquid nitrogen. The samples were stored in the refrigerator at −80 °C for the subsequent experiments.

### 2.4. Growth Performance Measurement

Equations below
Weight gain rate (WGR, %) = (*Wi* − *W*_0_)/*W*_0_ × 100%;
Specific growth rate (SGR, %/d) = (Ln *Wi* − Ln *W*_0_)/*t* × 100%;
Feed conversion ratio (FCR) = *Wf*/(*Wi* − *W*_0_);
Survival rate (SR, %) = *Ni*/*N*_0_ × 100%.
where *W*_0_ is the initial body mass of the fish (g), *Wi* is the final body mass (g), *Wf* is the feed intake (g), *N*_0_ is the initial number (tail), *Ni* is the final number (tail), and *t* is the number of days for the breeding experiment (d).

### 2.5. Enzymatic Assay

The digestive enzymes, antioxidant enzymes, and immune enzymes were assayed with corresponding commercial detection kits (C016-1-1, A054-1-1, A007-1-1, A001-3-2, A001-3-1, A060-2-1, Nanjing Jiancheng Bioengineering Institute, Nanjing, China). These digestive enzymes included α-amylase (AMS) and lipase (LPS), the antioxidant enzymes included catalase (CAT), superoxide dismutase (SOD), and the immune enzymes included acid phosphatase (ACP), and alkaline phosphatase (AKP). A bicinchoninic acid (BCA) protein assay kit (Beyotime Biotechnology, Haimen, China) was used to determine the protein concentration.

### 2.6. qPCR Analysis of Immune-Related Genes and Antioxidant Genes

The total RNA was extracted from intestinal cells by the TRIzol method, and RNA was examined by 0.1% agarose gel electrophoresis and a Nanodrop 2000 spectrophotometer (Thermo Fisher, Waltham, MA, USA). RNA was used as a template for reverse transcription to synthesize cDNA. Based on an ABI 7500 gene quantitative real-time detection system (Thermo Fisher, USA), relative expression of the immune genes interleukin-1beta (IL-1β), interleukin-10 (IL-10), Myeloid differentiation primary response gene 88 (MyD88), type I interferon (IFN1), toll-like receptor 2 (TLR-2), superoxide dismutase (SOD), catalase (CAT), and glutathione peroxidase 1 (GPX1) were determined by SYBR green I method. Specific primers for genes are demonstrated in Table 1. The reaction system for real-time qPCR was 20 μL, including 0.4 μL upstream and downstream primers (10 μmol/L), 6 μL cDNA template, 10 μL SYBR Mix I, and 3.2 μL diethylpyrocarbonate (DEPC) water. The program was 50 °C for 20 s, 95 °C for 7 min, followed by 40 cycles of 95 °C for 10 s and 60 °C for 30 s. The relative mRNA expression levels were detected by the 2^−△△CT^ method.

### 2.7. Infection Experiment of Aeromonas Hydrophila

After the sample collection, the remaining 20 fish in each group were used for the challenge experiment. Each fish was intraperitoneally injected with A. hydrophila suspension with a concentration of (3.3 × 10^7^) cfu/mL 100 μL in the experimental group, while the control group was injected intraperitoneally with 0.65% normal saline 100 μL. The death of fish in each group was observed and recorded, and the survival rate (SR) was calculated according to the following formula.
Survival rate (%) = Final survival mantissa/initial mantissa × 100

### 2.8. Intestinal Microbial Composition

The total DNA of microbes in the intestine was extracted directly with the E.Z.N.A. Stool DNA Kit (OMEGA, Stamford, CT, USA). Amplification of the hypervariable region (V4) of the bacterial 16S DNA gene was performed for sequencing analysis and species identification. The PCR products obtained by amplification were detected by 1% agarose gel electrophoresis, cut and recovered, and sequenced and analyzed on Novogene’s Illumina MiSeq platform (Tianjin, China) after accurate quantification. The method of sequencing and analysis referred to Yang et al. [17].

### 2.9. Statistical Analysis

All the data were presented as mean ± standard deviation (mean ± S.D.). Statistical analysis of experimental data was performed using SPSS 26.0 software and one-way ANOVA followed by LSD and Duncan’s test for multiple comparisons of significant differences between groups, with *p* < 0.05 considered the significant level.

## 3. Results

### 3.1. Growth Performance of Juvenile Turbot

The growth performance of juvenile turbot is shown in Table 2. After the culture experiment, there was no significant difference in survival rates of turbot among all groups (*p* > 0.05). The FBL, FBW, SGR, and FCR of the H1 and H2 groups were significantly higher than those of the C group (*p* < 0.05), and there was no significant difference between the two EPS groups. The WGR of the H2 group was the highest, which was significantly higher than those of the C and H1 groups (*p* < 0.05). Therefore, the addition of HMX2-EPS had a significant effect on the growth of juvenile turbot, while the addition concentration had little effect on the growth of turbot.

### 3.2. Digestive and Immune-Related Enzyme Activities in Juvenile Turbot

As shown in Table 3, the LPS activity in the H2 group was the highest and significantly higher than that in the H1 and C groups (*p* < 0.05). The AMS activity in the H1 and H2 groups was significantly higher than that of the C group (*p* < 0.05). In addition, the AKP and ACP activities of the groups H1 and H2 were significantly higher than that of the C group (*p* < 0.05), and there was no significant difference between the two groups (*p* > 0.05). The activity of CAT in the H1 and H2 groups was significantly higher than that in group C (*p* < 0.05), while SOD activity in the H2 group was the highest, and there was a significant difference between H1 and H2 groups, which was significantly higher than that in group C (*p* < 0.05). The results showed that the addition of HMX2-EPS to the diet could improve the digestive enzyme activity and the non-specific immune enzyme activity in the intestine of turbot.

### 3.3. Expression of Immune-Related Genes and Antioxidant Genes

As revealed in Figure 1, the addition of HMX2-EPS had no significant effect on the expression of TLR-2 and MyD88 (*p* > 0.05). However, the gene expression of 1L-1β and 1L-10 in H1and H2 groups was significantly higher than that in the C group (*p* < 0.05), and the expression of IFN1 in the H2 group was the highest and significantly higher than that in the H1 and C groups (*p* < 0.05). Compared with the control group, the gene expression of CAT, SOD, and GPX1 was upregulated in the HMX2-EPS groups (*p* < 0.05). It showed that the addition of large doses of HMX2-EPS could enhance non-specific immunity and antioxidant capacity.

### 3.4. Survival Rate after A. hydrophila Challenge

Different doses of HMX2-EPS could improve the survival rate of juvenile turbot after being poisoned by *A. hydrophila* (Figure 2). The highest survival rate was in the H2 group, which maintained 75% after 48 h, while the survival rate of juvenile turbot in the control group was only 20%.

### 3.5. Intestinal Microbiota Regulation by HMX2-EPS

#### 3.5.1. OUT Number and Alpha Diversity Analysis

A total of 60,352–68,279 effective sequences were obtained from the intestinal samples of juvenile turbot. Alpha diversity indices are presented in Table 4. Compared with the control group, the observed species, Shannon, Chao1, ACE, and PD whole tree indices, were significantly increased (*p* < 0.05) in H1 and H2 groups. However, there was no significant difference in the Simpson data among the three groups (*p* > 0.05). The results showed that HMX2-EPS supplementation could improve the abundance and evenness of microorganisms in the intestine.

#### 3.5.2. Beta Diversity Analysis

As shown in Figure 3A, the beta diversity analysis, according to the nonmetric multidimensional scaling (NMDS), showed that there was a difference between the C, H1, and H2 groups (stress = 0.019). The clustering results are presented at the phylum level (Figure 3B). H1 and H2 groups were clustered into a branch and then clustered with the C group into a branch. The results showed that the species composition structure of the H1 group and H2 group was similar, and the diversity of flora was higher than that of the C group.

#### 3.5.3. Microbial Composition and LefSe Analysis

At the phylum level (Figure 3B), *Proteobacteria* (20.01–42.19%), *Firmicutes* (17.71–28.02%), and *Bacteroidota* (11.13–23.2%) formed the core microbiota. *Proteobacteria* was the dominant phylum in the control group, which was higher than that in the H1 and H2 groups. Compared with the control group, the abundance of *Bacteroidota*, *Firmicutes*, and *Acidobacteriota* significantly increased in the H1 and H2 groups. At the genus level (Figure 3C), the abundance of potential probiotic-related genera, including *Bacteroides*, *Megamonas*, *Muribaculaceae,* and *Listeria* increased, and harmful bacteria such as *Ralstonia* and *Pseudomonas* decreased in the H1 and H2 groups.

To determine the difference in the abundance of species among populations, we performed the LEfSe analysis (Figure 4). We could see the analysis and the advantages of group C in phyla *Proteobacteria*, families *Psychromonadaceae*, *Pseudomonadaceae,* and *Methyloligellaceae*. The advantages of group H1 bacteria in phyla *Bacteroidota* and *Firmicutes*, families *Flavobacteriaceae*, *Streptococcaceae*, *Staphylococcaceae,* and *Enterococcaceae*. The advantages of group H2 bacteria in phyla *Firmicutes*, families *Bacillaceae* and *Lachnospiraceae*.

#### 3.5.4. Microbial Function Prediction

The abundance of functional categories by gut microbes at the second level (subfunction in each major type of metabolic pathway) is shown in Figure 5. Compared with the control group, the intestinal microorganisms in H1 and H2 groups were rich in enzyme families, immune systems, carbohydrate metabolism, and biosynthesis of other secondary metabolites. However, the abundance of some disease-related pathways was reduced, such as endocrine and metabolic diseases and infectious diseases.

## 4. Discussion

EPS is the medium of interaction between probiotics and host organisms [7]. Although the effects of probiotics in improving the growth performance, immune response, and regulating intestinal flora of fish have been extensively studied in the past few decades [18], studies on the effects of EPS isolated from *Lactiplantibacillus plantarum* on juvenile turbot have not been reported. At present, we have isolated a potential probiotic called *L. plantarum* HMX2, which had high EPS production. Therefore, this experiment was conducted to investigate the effects of the EPS produced by *L. plantarum* HMX2 on the growth performance, antioxidation, digestion, immune ability, and intestinal flora of *S. maximus*.

Compared with EPS from animals and plants, EPS produced by bacteria had many advantages, such as strong operability, high reproductive ability, and advanced performance [11]. Li et al. [19] found that the appropriate EPS produced by Antarctic bacteria *Pseudoalteromonas* sp. 3-3-1-2 could effectively increase the body weight and body length of *Pseudopleuronectes yokohamae*. In this study, the addition of HMX2-EPS to the feed significantly improved the growth performance of juvenile turbot, although no significant difference was observed when different concentrations of EPS were added.

After ingestion of EPS, the first target immune reaction site was the intestinal mucosa. The intestine was considered the largest digestive, absorption, immune, and endocrine organ in animals [20]. Therefore, we used the intestine of the turbot juvenile to conduct the related experiments. Digestive enzymes were important indicators of digestive capacity and played an important role in the growth and development of fish. According to different digestive objects, it could be roughly divided into lipase, amylase, protease, and cellulase [21]. The addition of *Bifidobacterium longum* A17 EPS to the diet could improve the activities of digestive enzymes AMS, LPS, and protease in goldfish intestines under cadmium stress [22]. In this study, HMX2-EPS could improve the activities of digestive enzymes AMS and LPS in the intestine of juvenile turbot, and 500 mg/kg of HMX2-EPS showed the best effect. The results illustrated that EPS had a promoting effect on the digestive system, which ensured the intestinal absorption of nutrients and thus achieved the effect of improving the growth performance of fish.

The antioxidant capacity of the organism’s defense system was closely related to health. Defense systems included both enzymatic and non-enzymatic systems, such as total antioxidant capacity (T-AOC), SOD, CAT, GPX, and glutathione peroxidase (GSH Px) belonged to the enzymatic system [23,24]. In this study, the addition of HMX2-EPS could improve the activity of SOD and CAT and promote the expression of antioxidant-related genes GPX1, CAT, and SOD in juvenile turbot. Chen et al. [25] found that the antioxidant capacity of *Cyprinus carpio L* was improved by the EPS from *Lactbacillu plantarum* HS-07 at different concentrations, and the activities of T-SOD, GSH Px, and CAT were significantly higher than those of the control group. Similar antioxidant capacity was demonstrated for EPS-9 from *Lactococcus lactis*, which significantly enhanced the activities of T-SOD, CAT, and GSH PX in the hepatopancreas and increased the concentrations of T-AOC and GSH in *Cyprinus carpio* [26]. The antioxidant activity of HMX2-EPS to improve the activity of antioxidant enzymes, as observed in this study, suggests the potential application of HMX2-EPS as an antioxidant agent to protect fish from body damage caused by oxidative stress.

ACP and AKP were not only important non-specific hydrolytic immune enzymes in the immune defense system but also important metabolic regulatory enzymes, which could be used as indicators to discriminate the body’s immune competence in aquatic animals [27]. The activities of AKP, ACP, SOD, and CAT in red tilapia were significantly improved by *Shewanella frigiimarina* W32-2 EPS, and the addition of 3 mg/mL was optimal [28]. EPS 20 # 3 and 3-3-1-2 produced by polar bacteria *Pseudoalteromonas paragorgicola* could significantly improve the non-specific immunity of turbot with significantly increased activities of AKP, ACP, T-SOD, and CAT [29]. In this study, the addition of high concentration HMX2-EPS could significantly improve the activities of AKP and ACP in the intestine, indicating that the EPS could improve the activity of immune enzymes in fish.

Most LAB and their EPS could protect against pathogens by improving the phagocytic ability of monocytes, promoting the proliferation of lymphocytes, and inducing cytokines to regulate the immune system, thereby improving the body’s immune defense ability [30]. Therefore, the detection of cytokines was often used to evaluate the immune response of the body. Zhang et al. [31] EPS from *Lactobacillus rhamnosus* LGG and *Lactobacillus casei* BL23 to the feed significantly increased the expression of pro-inflammatory factor (1L-6, TNF-a) and anti-inflammatory factor (1L-10) to improve the innate immunity of zebrafish. In this study, HMX2-EPS had no effect on the expression of TLR-2 and MyD88, indicating that HMX2-EPS did not activate the TLR-2/MyD88 pathway to induce inflammation. However, a high dose of HMX2-EPS significantly promoted the expression of 1L-1β, 1L-10, and IFN1. IFN1 is a crucial type I interferon for fish and provides the host with strong protection against pathogen infection. After pathogen infection was detected by the corresponding PRR, IFN was produced and secreted by multiple types of immune and non-immune cells, and the secreted IFN orchestrated innate and adaptive immunity through different mechanisms [32]. The results showed that HMX2-EPS could increase the secretion of inflammatory factors to trigger an inflammatory response, and it might enhance the immune response of turbot by regulating the IFN signal transduction pathway, thus promoting the immune response to invasive pathogens.

It was reported that the main pathogenic bacteria that could infect fish included *Vibrio anguillarum*, *Vibrio harveyi*, *Aeromonas hydrophila*, *Edwardia tarda,* and so on, which could cause symptoms such as body surface ulcers, ascites, and organ necrosis [33]. Wang [34] found that *Flavobacterium cloumnare* Pf1 EPS significantly improves the survival rate of grass carp infected by *F. cloumnare*. In this experiment, HMX2-EPS could improve the survival rate of juvenile turbot infected with *A. hydrophila*, perhaps because the high expression of IFN1 improves the disease resistance rate of turbot. The result showed that EPS could enhance innate immunity and had certain development value in aquatic production as an immune enhancer.

The intestine is a complex micro-ecosystem. The intestinal microbes of fish were closely related to the metabolism, nutrient absorption, growth and development, and immunity of the host, so maintaining the normal flora homeostasis in the intestine was essential for normal life activities [35]. The diversity of the gut microbiota correlated with the stability of the microbial community, and a decrease in diversity led to an increased risk of biological diseases [36]. In this study, the number of unique OUTs and alpha diversity indices in the EPS experimental groups was significantly increased compared with the control group, which indicated that the addition of HMX2-EPS in the diet significantly increased the abundance and evenness of intestinal microbiota of juvenile turbot. Beta diversity analysis found that EPS experimental group was significantly different from the control group, indicating that HMX2-EPS played a biological role and had an impact on the structure of intestinal flora. These results suggest that HMX2-EPS might be the key factor affecting intestinal flora.

Factors such as fish species, growth, habitat environment, feed species, and physiological characteristics of the gut all influence the amount and composition of the gut flora in fish, which could be both unified and interactive [35]. The gut microbiota is mainly composed of *Firmicutes*, *Bacteroidota*, *Proteobacteria,* and *Actinobacteriota*, which could effectively control the diversity and community structure of fish intestinal microorganisms [37]. There were two main ways for polysaccharides to regulate intestinal microorganisms. One was to regulate at the phylum level for *Firmicutes* and *Bacteroidota*. Another way was to promote the growth of beneficial bacteria and reduce the number of harmful bacteria, thus promoting intestinal health [38]. Both *Firmicutes* and *Bacteroidota* helped the host degrade many indigestible polysaccharides to provide energy and maintain the health of the animal intestines [38]. Generally, the imbalance of colonic flora and the increase of Proteobacteria abundance coexist. *Proteobacteria* could produce a large amount of endotoxin, which aggravates the structural destruction of intestinal epithelial tight junction and causes oxidative stress reaction and host metabolic disorder [39]. This study found that HMX2-EPS significantly increased the abundance of *Firmicutes* and *Bacteroidota* and decreased the abundance of *Proteobacteria*. *Lactobacillus rhamnosus* LGG-EPS could increase the abundance of *Firmicutes* and *Fusobacteria* and decrease the abundance of *Proteobacteria* and *Actinobacteria* in the zebrafish intestine to maintain the stability of the intestine flora structure [40], which was similar to our results. At the genus level, HMX2-EPS increased the reproduction of beneficial bacteria, such as *Bacteroides*, *Megamonas*, and *Muribaculaceae*, and decreased harmful bacteria, such as *Pseudomonas* and *Ralstonia*. Studies found that probiotics can colonize the intestine through the adhesion of EPS to intestinal epithelial cells. EPS could also promote probiotics’ growth, improve their stress resistance and prebiotic potential, and form a protective layer to help probiotics evade immune surveillance and promote the colonization of probiotics in the intestine [41]. The polysaccharide utilization sites of *Bacteroides* provided a major proteinaceous mechanism for the initiation of the metabolism of many carbohydrates [42]. *Megamonas* could prevent and treat inflammation-related diseases, such as digestive inflammation-related diseases, rheumatoid arthritis, and cardiovascular diseases [43]. *Muribaculaceae* was related to the intestinal mucosal immune system and had the function of promoting intestinal metabolism [10,44]. *Colwellia* and *Ralstonia* produced corresponding toxins that reduced host resistance [45,46]. These results illustrated that HMX2-EPS could promote more kinds of probiotics to colonize and reduce the adhesive space of harmful intestinal bacteria, which enabled the healthy development of intestinal flora in juvenile turbot.

The Tax4Fun was performed to predict the functional profiles of gut microbiota. The intestinal microorganisms of juvenile turbot in HMX2-EPS groups were rich in immune system, enzyme families, carbohydrate metabolism, and biosynthesis of other secondary metabolites. The functions predicted corresponded to the composition and abundance of the above described intestinal flora. Ma et al. [47] found that metabolism and endocrine functions were more abundant in the *Lactobacillus* HNUB20-EPS group than in the control group. These results confirmed that EPS could serve as a carbon source for gut microbes, providing energy to microbes and participating in a variety of life activities [48,49]. EPS had better functional and spatial characteristics, which contributed to the rapid degradation, utilization, and maintenance of intestinal homeostasis of various nutrients. It might also be one of the reasons for the enhanced growth performance and immune function of juvenile turbot.

## 5. Conclusions

In the present study, supplementation with HMX2-EPS significantly improved the growth performance and activities of antioxidant enzymes, digestive enzymes, and immune-related enzymes in vivo of juvenile turbot. HMX2-EPS also significantly improved the survival rate of turbot by enhancing the immune response through regulating the IFN signal transduction pathway. Moreover, HMX2-EPS could improve the diversity of intestinal microbiota and improve the function of gut microbes in metabolism and the immune system. Therefore, there was a positive effect of HMX2-EPS supplementation in the feed on the cultivation of turbot juveniles.

## Figures and Tables

**Figure 1 foods-12-02051-f001:**
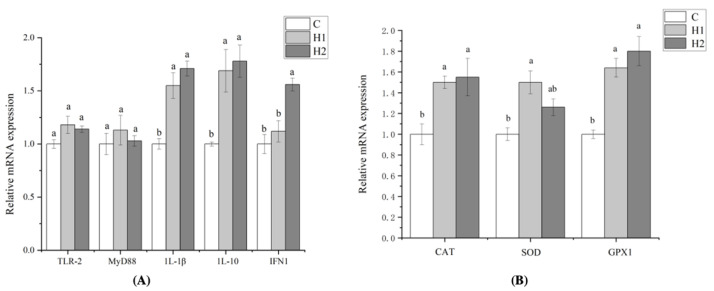
Effects of HMX2-EPS supplementation on the expression of immune-related genes in the intestines of juvenile turbot (**A**,**B**). C, 0 mg/kg; H1, 100 mg/kg; H2, 500 mg/kg. Different superscripts in each line indicate significant differences (*p* < 0.05).

**Figure 2 foods-12-02051-f002:**
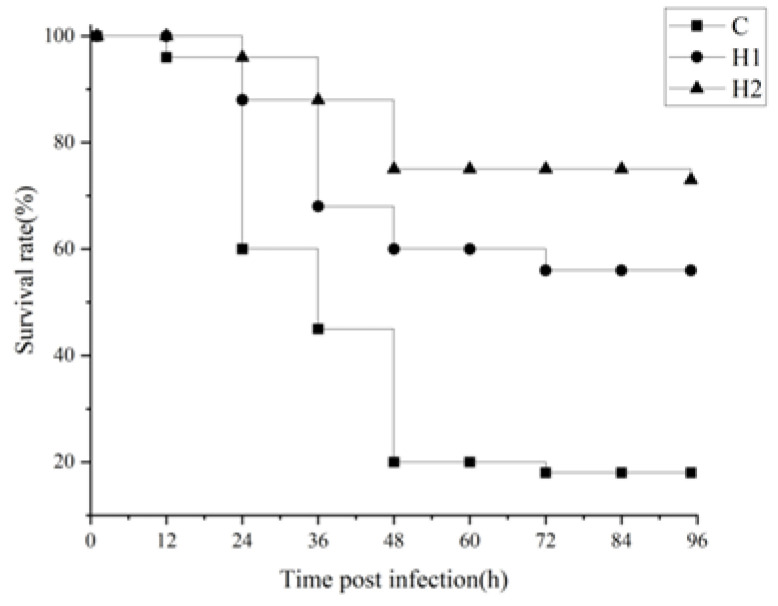
Effects of HMX2-EPS supplementation on the survival rate (%) of juvenile turbot challenged by *A. hydrophila*.

**Figure 3 foods-12-02051-f003:**
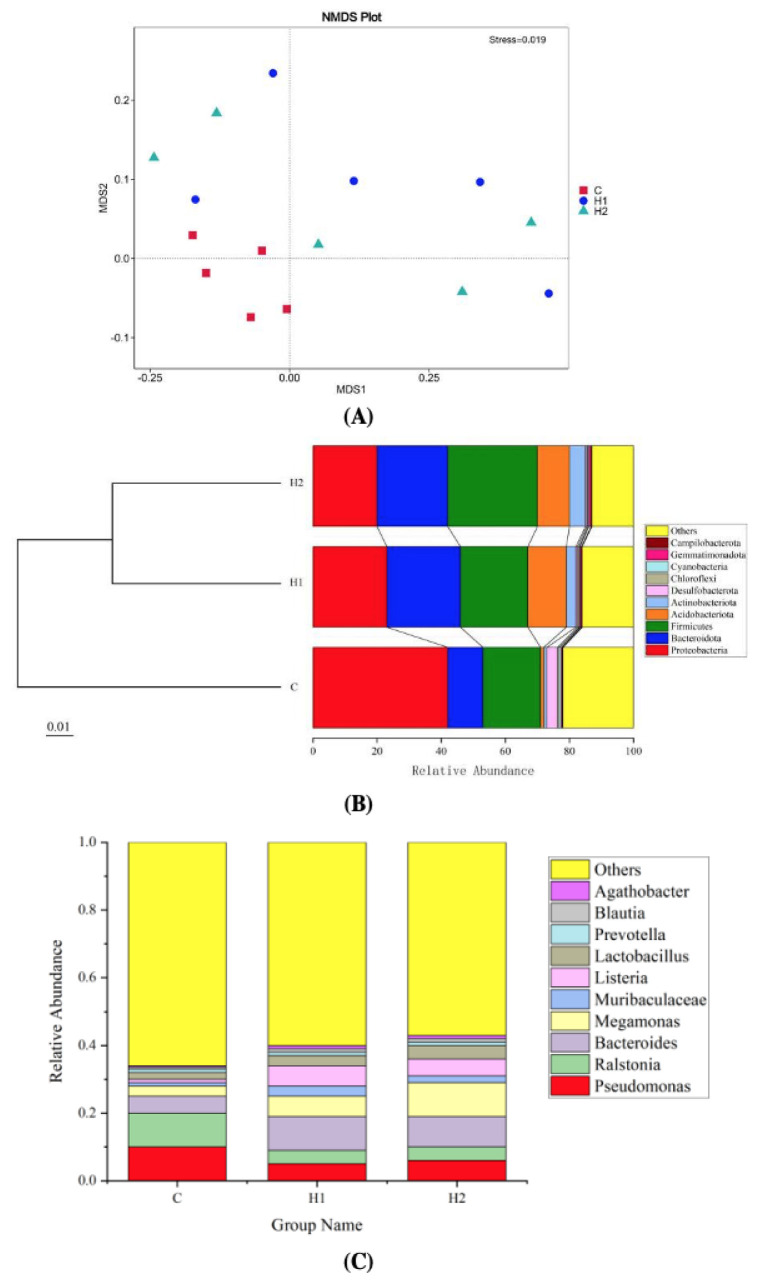
Nonmetric multidimensional scaling (NMDS) analysis of the intestinal bacterial community from juvenile turbot (**A**). Notes: NMDS can accurately reflect the degree of difference between samples when stress <0.2. Unweighted UniFrac distance matrix is used for UPGMA clustering analysis, and the clustering results are presented at the phylum level (**B**). Column charts of relative abundance of the top 10 gut microbiota at the genus level (**C**). C, 0 mg/kg; H1, 100 mg/kg; H2, 500 mg/kg.

**Figure 4 foods-12-02051-f004:**
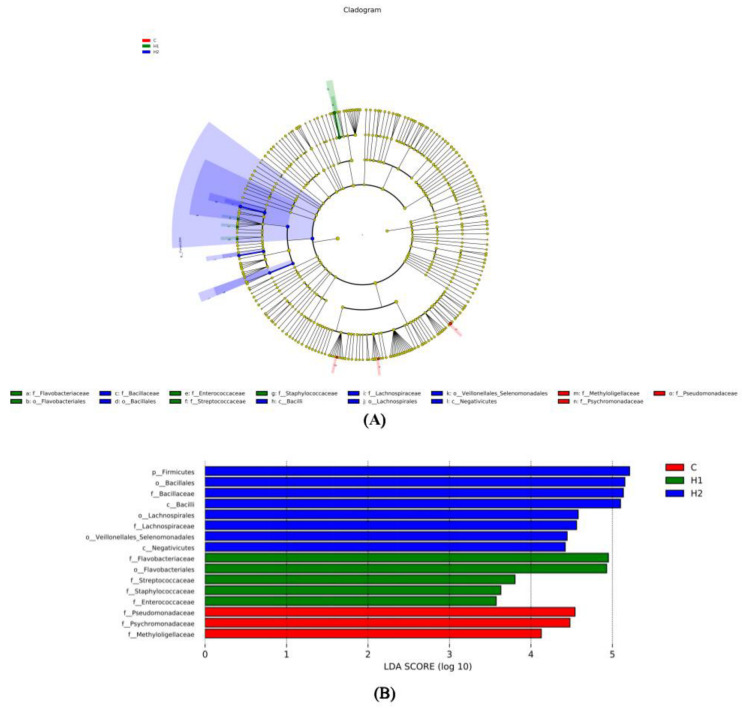
Linear discriminant analysis effect size (LEfSe) analysis of the gut microbiota fed with HMX2-EPS supplementation diets (*p* ≤ 0.05 and linear discriminant analysis (LDA) cut-off > 4) (*n* = 6). (**A**): Cladogram showing differences in the abundance of taxa; (**B**): linear discriminant analysis scores of the abundance of taxa. C, 0 mg/kg; H1, 100 mg/kg; H2, 500 mg/kg.

**Figure 5 foods-12-02051-f005:**
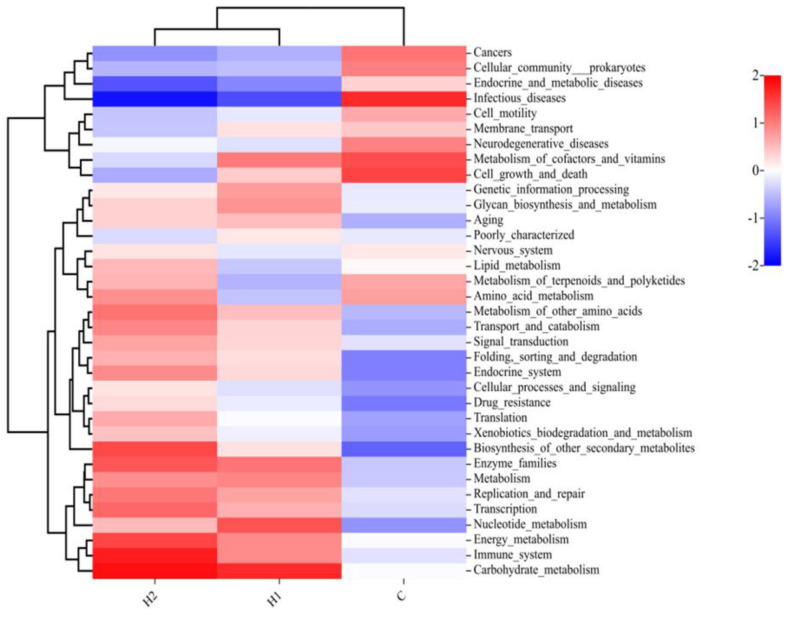
Heatmap of microbial functions at the second level (subfunction in each major type of metabolic pathway) of juvenile turbot. C, 0 mg/kg; H1, 100 mg/kg; H2, 500 mg/kg.

**Table 1 foods-12-02051-t001:** Primers of internal reference and target genes.

Gene	Primer Name	Sequence (5′->3′)	Genebank ID
β-actin	β-actin F	GCTGTCTTCCCTTCTATCGTCG	XM_035614477
	β-actin R	TCCATGTCATCCCAGTTGGTC	
IL-1β	IL-1β F	ATGGAGTGCAACATGAGCGA	XM_035640817
	IL-1β R	GAGCAGGTTTTCGTCCCTGA	
IL-10	IL-10 F	AGCTCAAGTCCGATGTCAGC	XM_035632547
	IL-10 R	TCAAGAGCTGGGTGATGCAC	
IFN-1	IFN-1 F	CCGACGGGCATTATGGGTAG	XM_035615472
	IFN-1 R	CGAGTATCCCTTCGTCCCAC	
MyD88	MyD88 F	CCCAATGGTAGCCCTGAGAT	XM_035618686
	MyD88 R	CATCTCGGTCGAACACACAC	
TLR-2	TLR-2 F	CTTCGAGCCAGGTAAACCC	XM_035606298
	TLR-2 R	AGAAAGACCAGGATCAGCACG	
SOD	SOD F	TATCAAGAGGCGCTGGCAAA	XM_035617519
	SOD R	GCTTCCATTAGCTCCCCCTG	
CAT	CAT F	CCAGAAACCCAGCCTCACTT	XM_035643471
	CAT R	GAAGGCACGGACCTGTGTAA	
GPX1	GPX1 F	TTCTGCCAAGGGACTCGTTG	XM_035631441
	GPX1 R	TCAAAGCCATTCCCTGGACG	

**Table 2 foods-12-02051-t002:** Effects of HMX2-EPS supplementation on the growth performance of juvenile turbot. IBL, initial body length; IBW, initial body weight; FBL, final body length; FBW, final body weight; WGR, weight gain rate; SGR, specific growth rate; FCR, feed conversion ratio; SR, survival rate. C, 0 mg/kg; H1, 100 mg/kg; H2, 500 mg/kg. Different superscripts in each line indicate significant differences (*p* < 0.05).

Items	Treatments		
	C	H1	H2
IBL (cm)	5.94 ± 0.07	5.85 ± 0.12	5.93 ± 0.14
IBW (g)	6.11 ± 0.60	6.15 ± 0.45	6.16 ± 0.31
FBL (cm)	7.54 ± 0.21 ^b^	8.36 ± 0.11 ^a^	8.44 ± 0.07 ^a^
FBW (g)	21.07 ± 0.34 ^b^	23.29 ± 0.59 ^a^	24.15 ± 0.38 ^a^
WGR (%)	244.84 ± 3.72 ^c^	278.70 ± 8.76 ^b^	292.05 ± 6.68 ^a^
SGR (%/d)	2.95 ± 0.1 ^b^	3.17 ± 0.09 ^a^	3.25 ± 0.11 ^a^
FCR	1.04 ± 0.08 ^b^	1.17 ± 0.04 ^a^	1.14 ± 0.05 ^a^
SR (%)	98.33 ± 2.36	100.00	98.33 ± 2.36

**Table 3 foods-12-02051-t003:** Effects of HMX2-EPS supplementation on the digestive enzyme activities and immune-related enzyme activities of juvenile turbot. C, 0 mg/kg; H1, 100 mg/kg; H2, 500 mg/kg. Different superscripts in each line indicate significant differences (*p* < 0.05).

Items	Treatments		
	C	H1	H2
LPS (U/gprot)	23.68 ± 2.11 ^b^	25.51 ± 1.23 ^b^	35.13 ± 4.76 ^a^
AMS (U/gprot)	0.42 ± 0.04 ^b^	0.59 ± 0.06 ^a^	0.64 ± 0.02 ^a^
AKP (U/100mL)	79.23 ± 5.92 ^b^	104.76 ± 10.64 ^a^	98.23 ± 3.87 ^a^
ACP (U/100mL)	90.58 ± 3.44 ^b^	111.29 ± 13.13 ^a^	108.32 ± 9.22 ^a^
SOD (U/mL)	405.01 ± 18.98 ^c^	533.12 ± 13.13 ^b^	655.94 ± 24.22 ^a^
CAT (U/mL)	13.39 ± 5.44 ^b^	53.56 ± 9.33 ^a^	48.38 ± 8.14 ^a^

**Table 4 foods-12-02051-t004:** Effects of HMX2-EPS supplementation on the α diversity of the intestinal flora of juvenile turbot. Different superscripts in each line indicate significant differences (*p* < 0.05).

Items	Treatments		
	C	H1	H2
observed_species	990.91 ± 4.24 ^b^	1297.26 ± 99.15 ^a^	1406.81 ± 136.12 ^a^
Shannon	6.88 ± 0.61 ^b^	8.38 ± 0.12 ^a^	9.21 ± 0.81 ^a^
Simpson	0.91 ± 0.13	0.92 ± 0.67	0.95 ± 0.62
Chao1	1310.50 ± 14.33 ^b^	1668.27 ± 16.14 ^a^	1848.50 ± 26.17 ^a^
ACE	810.18 ± 8.18 ^b^	1863.23 ± 19.48 ^a^	1987.27 ± 12.42 ^a^
PD_whole_tree	92.51 ± 1.27 ^b^	154.18 ± 1.36 ^a^	174.79 ± 2.88 ^a^

## Data Availability

The data presented in this study are available on request from the corresponding author.

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
