# Peer review of "Effects of the Exopolysaccharide from Lactiplantibacillus plantarum HMX2 on the Growth Performance, Immune Response, and Intestinal Microbiota of Juvenile Turbot, Scophthalmus maximus"

_foods, 2023, doi:10.3390/foods12102051_

Round 1

Reviewer 1 Report

This is a very good quality scientific paper well designed, well described and well discussed. I have serious doubts about this paper because it's more zootechnique paper than food science, this paper discusses growth of fish, but not fish meat as a raw material. Some of the ratings which I indicated is a result of this improper matching.

It is necessary to provide more information in the introduction: why this kind of fish, why this kind of probiotic bacteria, how about other similar studies, where they carried out or this is the first one published?

Line 15 and 68: it's not sauerkrait but sauerkraut

Change word microflora into microbiota. Whenever it is possible, we try to avoid the word microflora because flora means plants which is not correct in this case

Author Response

Dear Editor

FOODS

We sincerely thank the reviewers for their valuable comments and suggestions. We have revised the manuscript according to their valuable comments. We have addressed their comments point by point. The revisions made to the revise manuscript can be seen in track changes.

We hope that now the article would be accepted for publication.

Reviewer: #1

This is a very good quality scientific paper well designed, well described and well discussed. I have serious doubts about this paper because it's more zootechnique paper than food science, this paper discusses growth of fish, but not fish meat as a raw material. Some of the ratings which I indicated is a result of this improper matching. 

Response: Because the topic of this Special Issue is Functional Properties of Food Source Probiotics, our research focuses on the effects and functions of exopolysaccharide from Lactiplantibacillus plantarum derived from northeast sauerkraut as feed for juvenile turbot. Exopolysaccharides are important secondary metabolites of lactic acid bacteria and are widely studied for applications in food including feed. Scophthalmus maximus is an economically important species of flounder known as turbot. Its meat is delicious and rich in nutritional value, widely cultivated in Europe and China. At present, there are few research has been reported on food derived lactic acid bacteria exopolysaccharides as feed additives for aquatic animals. Our results indicated that HMX2-EPS supplementation in the diet could promote the growth, improve antioxidant activity, digestive capacity, and immunity capacity, and actively regulate the intestinal microflora of juvenile turbot. This study might provide technical support and scientific basis for the application of L. plantarum in aquatic feed. So, we chose to submit the paper to this special issue of the Journal.

It is necessary to provide more information in the introduction: why this kind of fish, why this kind of probiotic bacteria, how about other similar studies, where they carried out or this is the first one published?

Response: Thank you for the constructive comments on the manuscript. We added the description at L66-L79.

Line 15 and 68: it's not sauerkrait but sauerkraut

Response: Thank you for your comment. It has been corrected.

Change word microflora into microbiota. Whenever it is possible, we try to avoid the word microflora because flora means plants which is not correct in this case

Response: Thank you for your comment. It has been corrected.

Reviewer 2 Report

see attachment!

Author Response

Dear Editor

FOODS

We sincerely thank the reviewers for their valuable comments and suggestions. We have revised the manuscript according to their valuable comments. We have addressed their comments point by point. The revisions made to the revise manuscript can be seen in track changes.

We hope that now the article would be accepted for publication.

Reviewer: #2

Write at which level of HMX2-EPS was significant.

Response: Thank you for the constructive comments on the manuscript. We added it at L25-L26.

Need of the project should be raised more effectively.

Response: Thank you very much. We added the description at L66-L79.

How sample was calculated? Which experimental design was followed?

Response: The experiment set up 300 fish and divided them into 3 treatment groups, with each treatment group repeated 4 times. The experimental design was followed by

Liu (2021), and each group of 100 fish can more intuitively monitor survival rates.

Liu Yingjie. Effects of Yeast Polysaccharides on growth performance and intestinal digestive enzyme activities of carp[J]. Chinese Journal of Traditional Veterinary Science. 2021(7), 263.

Add mathematical model for better illustration of data analysis. Less and equal to. Why author’s selected DMR test as post hoc, any specific reason? LSD test can be more effective in this case.

Response: Thank you for the constructive suggestion. We have consulted many similar literature and adjusted for a better description at L196-L199. LSD and Duncan's test are more concise and convenient to use together.

Add actual p-value in all tables.

Response: Thanks. We have checked the p-value in the table to ensure they are accurate.

Check regression analysis by orthogonal polynomial contrast to evaluate linear and quadratic trend.

Response: Thank you very much. We have checked regression analysis carefully.

This section needs attention add logical reasoning of each result before discuss with previous studies.

Response: Thank you for the constructive comments on the manuscript. We have adapted the sequence of results and discussion in sections.

Conclusions should be brief and concise.

Response: Thank you very much. We have revised the conclusions.

Reviewer 3 Report

The MS is worth of interest and it is well-written and explicative. However, I have some doubts about the fitting of the MS to the Journal. there are different journals of MPDI dedicated to Aquaculture studies. Why dis you decide to send the MS to Foods?

In addition to this, there are a few points that should be clarified before fully considered for publication:

1) it is not clear why and how the two concentrations of HMX2-EPS were selected. Have you made a pre-trial before selecting the concentrations? And Why only these two concentrations were selected? Please give details

2) why did you select to divide 300 fish into 3 groups instead of considering more and different concentrations of HMX2-EPS and putting a fewer number of fish for groups?

3) details about the ethical methodologies used to sacrifice the fish are missed.

Please find all the detailed comments in the pdf file attached.

Author Response

Dear Editor

FOODS

We sincerely thank the reviewers for their valuable comments and suggestions. We have revised the manuscript according to their valuable comments. We have addressed their comments point by point. The revisions made to the revise manuscript can be seen in track changes.

We hope that now the article would be accepted for publication.

Reviewer: #3

The MS is worth of interest and it is well-written and explicative. However, I have some doubts about the fitting of the MS to the Journal. there are different journals of MPDI dedicated to Aquaculture studies. Why dis you decide to send the MS to Foods?

Response: Thank you for your comment. Because the topic of this Special Issue is Functional Properties of Food Source Probiotics, our research focuses on the effects and functions of exopolysaccharide from Lactiplantibacillus plantarum derived from northeast sauerkraut as feed for juvenile turbot. Exopolysaccharides are important secondary metabolites of lactic acid bacteria and are widely studied for applications in food including feed. Scophthalmus maximus is an economically important species of flounder known as turbot. Its meat is delicious and rich in nutritional value, widely cultivated in Europe and China. At present, there are few research has been reported on food derived lactic acid bacteria exopolysaccharides as feed additives for aquatic animals. Our results indicated that HMX2-EPS supplementation in the diet could promote the growth, improve antioxidant activity, digestive capacity, and immunity capacity, and actively regulate the intestinal microflora of juvenile turbot. This study might provide technical support and scientific basis for the application of L. plantarum in aquatic feed. So we chose to submit the paper to this special issue of the Journal.

In addition to this, there are a few points that should be clarified before fully considered for publication:

  • it is not clear why and how the two concentrations of HMX2-EPS were selected. Have you made a pre-trial before selecting the concentrations? And Why only these two concentrations were selected? Please give details

Response: Thank you for your comment. Based on the results of Lu et al (2021) and Chen Juan Li (2022), 100 mg/kg and 500 mg/kg of HMX2-EPS were added to the basal diet. We did pre-experiments and these two concentrations were most appropriate.

Lu Wenqi, Liang Ying, Zhai Shaowei, et al. Preliminary study on effect of adding lentinan in diet of European eel (Anguilla anguilla).Feed Research. 2021(21):71-74.

Chen Jian, Dai Bingtao, Wang Hongming, et al. Effects of adding β-glucan to feed on the growth performance,immune indexes,transcriptome and intestinal flora ofEpinephelus fuscoguttatus ♀ × Epinephelus lanceolatus. Journal of Southern Agriculture. 2022,53(5): 1434-1447

  • why did you select to divide 300 fish into 3 groups instead of considering more and different concentrations of HMX2-EPS and putting a fewer number of fish for groups?

Response: Thank you for your comment. The experiment set up 300 fish and divided them into 3 treatment groups, with each treatment group repeated 4 times. The experimental design was followed by

Liu (2021), and each group of 100 fish can more intuitively monitor survival rates.

Liu Yingjie. Effects of Yeast Polysaccharides on growth performance and intestinal digestive enzyme activities of carp[J]. Chinese Journal of Traditional Veterinary Science. 2021(7), 263.

  • details about the ethical methodologies used to sacrifice the fish are missed.

Response: Thank you for your comment. We have mentioned the relevant in the Ethical Approval at L604-L607.

Please find all the detailed comments in the pdf file attached.

This sentence is confused. Consider: In this study, the economically important flatfish turbot (Scophthalmus maximus) was used as the experimental species[12].

Response: Thank you for the constructive comments. We have modified this section of the description.

Align the text.

Response: Thank you for your comment. The text has been aligned in the revised manuscript.

TCA.

Response: Thank you for your comment .It has been corrected.

Groups were

Response: Thank you for your comment .It has been corrected.

It is not clear how these two concentration were selected. Have you before made a pre-trial to selected the concentration? Why only these two concentration were selected? Please give details.

Response: Based on the results of Lu et al (2021) and Chen Juan Li (2022), 100 mg/kg and 500 mg/kg of HMX2-EPS were added to the basal diet. We did pre experiments and these two concentrations were most appropriate.

Lu Wenqi, Liang Ying, Zhai Shaowei, et al. Preliminary study on effect of adding lentinan in diet of European eel (Anguilla anguilla).Feed Research. 2021(21):71-74.

Chen Jian, Dai Bingtao, Wang Hongming, et al. Effects of adding β-glucan to feed on the growth performance,immune indexes,transcriptome and intestinal flora ofEpinephelus fuscoguttatus ♀ × Epinephelus lanceolatus. Journal of Southern Agriculture. 2022,53(5): 1434-1447

How? Please give details.

Response: Thank you for your comment. We have provided additional details in the revised manuscript please check line 116-117.

Please give details about the ethic methodologies used to sacrify the fish.

Response: Thank you for your comment. We have provided the relevant in the Ethical Approval at L604-L607.

Please double check the format.

Response: Thank you for your comment. We have revised the manuscript according to the Journal format.

This sentence sounds too assumptive. There are other different bacteria that are also known to be pathogenic for fish so please rephrase the sentence accordingly.

Response: Thank you for your comment, it has been corrected.

Fish, is

Response: Thank you for your comment, it has been corrected.

Round 2

Reviewer 2 Report

Good Paper